# Exposure to Chinese Famine in Fetal Life and the Risk of Dysglycemiain Adulthood

**DOI:** 10.3390/ijerph17072210

**Published:** 2020-03-25

**Authors:** Yan Zhang, Chao Song, Meng Wang, Weiyan Gong, Yanning Ma, Zheng Chen, Ganyu Feng, Rui Wang, Hongyun Fang, Jing Fan, Ailing Liu

**Affiliations:** National Institute for Nutrition and Health, Chinese Center for Disease Control and Prevention, 27 Nanwei Road, Xicheng District, Beijing 100050, China

**Keywords:** famine, type 2 diabetes, fetal life

## Abstract

Undernutrition in early life may have a long consequence of type 2 diabetes in adulthood. The current study was aimed to explore the association between famine exposure in fetal life during China’s Great Famine (1959–1961) and dysglycemia in adulthood. The cross-sectional data from 7830 adults from the 2010–2012 China National Nutrition and Health Surveillance was utilized. Participants who were born between 1960 and 1961 were selected as the exposed group, while the participants who were born in 1963 were selected as the unexposed group. Logistic regression was utilized to examine the relationship between fetal famine exposure and dysglycemia in adulthood. The prevalence of type 2 diabetes in the exposed and control group was 6.4% and 5.1%, respectively, and the risk of type 2 diabetes in the exposed group was 1.23 times higher than that of the control group (95%CI, 1.01–1.50; *P* = 0.042) in adulthood, and 1.40 times in the severely affected area (95%CI, 1.11–1.76; *P* = 0.004). The fasting plasma glucose of the exposed group was higher than that of the control group, which was only found in the severely affected area (*P* = 0.014) and females (*P* = 0.037). The association between famine and impaired fasting glucose was observed only in females (OR 1.31, 95%CI, 1.01–1.70; *P* = 0.040). Our results suggested that fetal exposure to Chinese famine increased the risk of dysglycemia in adulthood. This association was stronger in the severely affected area and females.

## 1. Introduction

According to the World Health Organization (WHO), an estimated 422 million adults were living with diabetes in 2014, compared to 108 million in 1980 worldwide. Over the past decade, diabetes prevalence has risen faster in low- and middle-income countries than in high-income countries [1]. Diabetes and complications caused by diabetes can bring about a substantial economic loss to patients and their families [2,3] and to the country. Diabetes is becoming a significant economic burden and a serious public health problem. Generally, common non-communicable chronic diseases, including diabetes, cardiovascular disease, fatty liver diseases, have been recognized as “rich diseases”, which are affected by affluent energy diet or physical inactivity [4]. However, increasing evidence supports that nutrition status in early life has a long consequence of chronic diseases in adults [5,6,7].

Barker [8,9] postulated that adaptations in response to fetal undernutrition led to metabolic and structural changes, which were beneficial for early survival, but might increase the risk of common diseases such as type 2 diabetes in adulthood. Due to ethical reasons, natural famine provided the best opportunity to carry quasi-experimental research on testing the hypothesis in humans. Different periods of famine around the world have been used to explore the association of early life undernutrition with the risk of diabetes. A meta-analysis has indicated that famine exposure during early life, especially fetal-infant exposure, may increase the risk of type 2 diabetes in adulthood [10].

The Chinese famine, documented as a “three-year natural disaster” in the Chinese literature, lasted from 1959 to 1961, and caused more than 30 million deaths and 33 million fertility losses [11]. Famine is considered the largest and most severe famine during the 20 century. Most babies born at this famine suffer severe fetal undernutrition for the sharp decrease of the grain output between 1959 and 1962 [12]. They are now a middle-aged (around 50 years old) population, who are very susceptible to diabetes. Fetal exposure to the Chinese famine has been associated with risks of overweight, hyperglycemia, and diabetes in adulthood [13,14,15]. However, there is a lack of national data to examine the association between fetal exposure to Chinese famine and adult type 2 diabetes.

Therefore, the data from the China National Nutrition and Health Surveillance (CNNHS), 2010–2012 were utilized to identify the associations of famine exposures in fetal life with risks of type 2 diabetes in adulthood.

## 2. Materials and Methods

### 2.1. Study Design

The China National Nutrition and Health Surveillance (CNNHS), 2010–2012 was a national representative cross-sectional study conducted by the National Institute of Nutrition and Health, the Chinese Center for Disease Control and Prevention, to assess the nutrition and health of Chinese civilians. According to the level and type of economic development, China’s county-level administrative units (include districts and county) in the mainland were divided into four strata: large cities, small to medium cities, general rural areas, and poor rural areas. A total of 150 surveillance sites (districts/counties) were selected, including 34 large cities, 41 small to medium cities, 45 general rural areas, and 30 poor rural areas. From each of the surveillance point, six residential villages/communities were randomly selected; and 75 households were then randomly sampled from each village/community [16].

The Chinese famine lasted for three years in 1959–1961; therefore, born between 1 January 1960 and 31 December 1961 were selected as the exposed group, and those born between 1 January 1963 and 31 December 1964 were selected as the control group. The participants who met one of the following criteria were excluded in this study: (1) no fasting blood glucose data or not participate in oral glucose tolerance test; (2) suffering the liver or kidney disease, or cancer; (3) had been diagnosed as type 2 diabetes and had changed the lifestyle. The total sample size was 7830 people (4081 in the exposed group, 3749 in the control group). Chinese famine was a national affected event, but the degrees of the famine severity were varied from the regions. Referred to the previous study [15], we divide the famine cohort into moderately and severely famine-affected areas. Briefly, famine severity was defined by the percentage increase in mortality rate from the average mortality rate for 1956–1958 to the highest mortality rate over 1959–1962. And the 50% was the cutoff value: regions that had an equal or higher rate than this cutoff were categorized as severely affected famine areas, and otherwise as moderately affected famine areas [17]. Among the exposed group, 1896 (47.7%) participants were from the moderate famine area, and 2079 (52.3%) participants were from severe famine areas. The protocol of the 2010–2012 CNNHS was approved by the Ethical Committee of the National Institute for Nutrition and Health, Chinese Center for Disease Control and Prevention (No. 2013-010). Signed consent forms were obtained from all participants.

### 2.2. Data Collection

The fasting venous blood samples were taken from each participant in the morning after 10–14 h of overnight fasting. Fasting plasma glucose (FPG) concentration was measured using the glucose oxidize enzymatic method within 3 h of blood collection. Participants without known diabetes were required to take a 75 g oral glucose loads and 2 h later, venous blood samples were collected to deter-mine 2-h plasma glucose concentrations.

Information about demographic characteristics, smoking, physical exercise, sedentary behaviors were collected using questionnaires by trained investigators. Educational levels were classified as primary school and low, junior middle school, senior high school, and above. The annual family income per capita was used to assess the economic status of participants, and according to by the per capita annual income of urban and rural households in 2011, the economic status was divided into three levels: low income (<20000 RMB), middle income (20000–40000 RMB) and high income (>40000 RMB). Smoking and drinking were coded as “yes” or “no”. Physical exercise in leisure time was coded as “yes” or “no”. The sedentary time was defined as time spent sitting or lying in leisure time.

Height was measured in bare feet to 0.1 cm. Fasting body weight was measured in the morning to the nearest 0.1 kg, with participants wearing lightweight clothing. Body Mass Index (BMI) was calculated as the weight in kilograms divided by height in meters squared (kg/m^2^).

The dietary intake was measured by a validated semiquantitative food frequency questionnaire (FFQ) and a 24-h dietary recall method. FFQ collected the consumption frequency and weight of common foods (118 items in 11 food categories) in the past 1 year. The whole food intake during the last three consecutive days (two weekdays and one weekend day) were collected by 24-h dietary recall method. The average daily intake of each food and food category were then calculated [16]. Previous studies had indicated associations of whole cereal, beans, livestock, and poultry intakes with diabetes [18,19,20]. Therefore, the present study involved the intake of whole cereal and beans and the intake of livestock and poultry as confounders, and the other information on the dietary intake was not presented. The whole cereal and beans intake level was divided into insufficient (less than 40g per day), sufficient (more than or equal to 40g and less than or equal to 75g per day), and very sufficient (more than 75 g per day) based on the “Dietary guideline for Chinese residents (2016)” [21]. Livestock and poultry intake level were divided into insufficient (less than 50 g per day), sufficient (more than or equal to 50 g and less than or equal to 150 g per day), and excessive (more than 150 g per day).

### 2.3. Related Definitions

Criteria proposed by WHO and International Diabetes Federation (IDF) 1999 on Diabetes Mellitus was used [22,23]. Type 2 diabetes was defined as an Fasting plasma glucose (FPG) ≥ 7.0 mmol/l and 2-h plasma glucose ≥ 11.0 mmol/l and a previous clinical diagnosis of type 2 diabetes. Impaired fasting glucose (IFG) was defined as an FPG ≥ 6.1 mmol/L and 2-h plasma glucose < 7.8 mmol/L. Impaired glucose tolerance (IGT) was defined as an FPG < 7.0 mmol/L, 2-h plasma glucose ≥ 7.8 mmol/L, and < 11.1 mmol/L.

### 2.4. Statistical Analysis

The SAS 9.4 (SAS Institute, Cary, NC) statistical program was used to analyze the data. Two-tailed *P* values < 0.05 were considered statistically significant. Shapiro–Wilk test was used to test the normality of the distribution of the data. Age, BMI, and FBG were normal distributed (*P* > 0.05) and were presented as mean ± standard deviation (SD). Sedentary time was skew distributed and was presented as median (P25, P75). N (%) was presented for categorical variables, including education and economic status, smoking, drinking, three primary endpoints, etc. To compare the difference in the basic information between those two groups, the Student’s t-test and Pearson’s chi-square tests were utilized for continuous variables and categorical variables, respectively. To examine the association of fetal famine exposure with the fasting plasma glucose and risks of dysglycemia, four multiple linear or logistic regression models were used. Model 1 was adjusted for gender, economic status, and education level. Model 2 was adjusted for the variables in model1 and smoking, drinking, physical exercise, sedentary time. Model 3 was adjusted for the variables in model 2 and dietary factors, including whole cereal and beans intake level, livestock and poultry intake level. Model 4 was adjusted for the variables in model 3 and BMI. Some, but not all, the relevant literature often report results with specified gender or area. We then run the four models on samples stratified by gender and severity of famine without the stratified variable.

## 3. Results

A total of 7830 participants (57.6% women) were included in the present study, with the age of 49.7 ± 1.5 years. The general characteristic of participants between the exposed and unexposed groups is shown in Table 1. Except for the age, economic status, education level, whole cereal, and beans intake level, the difference of basic characteristics between these two groups was not significant. The means of FPG in the exposed group and control groups were 5.4 mmol/L and 5.3 mmol/L, respectively. There was no difference in FPG between the two groups (*P* = 0.146). The prevalence of type 2 diabetes in the exposed group was higher than the control group (6.4% vs. 5.1%, *P* = 0.016). The prevalence of Impaired glucose tolerance (IGT) in the two groups was 5.3% and 5.0%, while the prevalence of IFG was 7.0% and 6.4%. There was no difference in the prevalence of IGT and IFG between these two groups.

Table 2 presents the associations of fetal famine exposure with FPG and the risk of dysglycemia in adulthood. After adjusting for gender, economic status, and education level, fetal famine exposure was associated with an increased risk of type 2 diabetes (*P* = 0.030). Additional adjustment for smoking, drinking, physical exercises and sedentary time, dietary factors, and BMI, the risk of type 2 diabetes in the exposed group is still higher than the control group, the odds ratio (OR) was 1.23 (95%CI, 1.01–1.50; *P* = 0.042). However, no increased concentration of fasting plasma glucose and risk of the IFG and IGT were observed in the four adjusted models.

The stratified results (Table 3 and Table 4) showed that the association of fetal famine exposure and glucose metabolism was stronger in severe famine-affected areas and females. In severe famine-affected areas, the fasting plasma glucose was 0.08mmol/L higher than the control group (P = 0.014), and the risk of type 2 diabetes was 1.40 times higher than that of the control group (95%CI, 1.11–1.76; P = 0.004) after adjusting for all of the confounding factors. In females, the increased fasting plasma glucose was significantly associated with fetal famine exposure after adjusting for all of the confounding factors (P = 0.037). The prevalence of impaired fasting glucose was significantly increased in the exposed group compared to the control group (OR, 1.31; 95%CI, 1.01–1.70; P = 0.040) after adjusting for confounding factors. There was no association of fetal famine with FPG and dysglycemia in moderate famine affected area and male.

## 4. Discussion

In the current study with a large sample of Chinese adults, we found that famine exposure in fetal life was associated with increased risks of type 2 diabetes in adulthood. This finding was found in severe famine-affected areas (not in moderately affected areas). In other previous famine studies in China, they also showed that exposure to Chinese famine during early life increased the risk of developing diabetes in adulthood. Wang et al. use the data from a 2014 cross-sectional survey, which was conducted in multi provinces of China, demonstrated that both prenatal and postnatal life might be important time windows for determining the future risk of diabetes [14,24]. Li et al. [25] showed that prenatal exposure to famine significantly increases the risk of hyperglycemia in 2 consecutive generations. Wang et al. [26] reported that individuals who experienced famine in childhood had increased diabetes risk in adulthood. These findings suggested that nutrition in fetal life and childhood may play an important role in the prevention and control of diabetes risk in later life. The current study confirmed that malnutrition during the prenatal period might have a long-term effect on the development of type 2 diabetes.

Significant famine effects also have been found in other countries. Lumey et al. [27] found a dose-response relation between Ukrainian famine severity during prenatal development and odds of type 2 diabetes in later life. Keinan et al. [28] focused on the unique population of Holocaust survivors who live in Europe during World War II when they were fetuses, infants or children, and found the association between early-life exposures and long-term chronic conditions. The long-term health consequence of the Dutch famine (‘Hunger winter’) is the most comprehensively studied up to now. Both Abeelen et al. [29] and Portrait et al. [30] indicated that a short period of severe undernutrition during childhood or young adolescence was associated with an increased risk of type 2 diabetes in adult life.

The mechanism of the relationship between fetal famine exposure and risk of type 2 diabetes in adulthood was still unclear. In the previous study, it was concluded that people who have a nutritionally rich environment in later life tended to have a higher risk of hyperglycemia in adulthood [30]. These may indicate that if nutrition status in adulthood mismatch with the nutrition in fetal life, there might be an increase in the risk of diabetes in adult life [31]. In addition, animal models of intrauterine growth restricted (IUGR) have provided a cure for understanding the famine effects. All of the rodent and sheep models have showed that fetal growth restriction in early life could result in dysfunction of beta-cells, including a reduction in beta cell mass, lower proliferation of beta cells, and impaired insulin secretion, which ultimately develops into glucose intolerance and fasting hyperglycemia in later life [32]. Furthermore, epigenetic changes in early life seem to play a critical role in the development of type 2 diabetes. Heijmans et al. [33] had reported that preconception exposure to famine during the Dutch Hunger Winter was associated with lower methylation of the IGF2. And epigenetic dysregulation is associated with several components that contribute to type 2 diabetes risk, including altered feeding behavior, insulin secretion, and insulin action [34]. Moreover, there was animal evidence shows that malnutrition in fetal life might affect appetite, feeding behavior, and adiposity in adulthood [35].

The current study found that fetal famine was associated with type 2 diabetes by not with two pre-diabetes outcomes (IGT and IFG) in adulthood. Most individuals with IGT and IFG eventually develop diabetes. These results indicated that famine might be a factor, which aggravates the development of diabetes instead of the independent risk factor.

However, the results of this current study show that the association between fetal famine and IFG had gender-difference (effects on impaired fasting glucose only observed in women). The gender-specify association was also found in other studies [26,30]. These findings could be explained by several reasons. First, there was animal evidence showing that prenatal caloric restriction female rats had an increase in caloric intake and compulsive feeding in adulthood compared to male rats [36], which might increase the risk of obesity. Obesity is regarded as a risk factor for diabetes and would be the intermediate factor on fetal famine and type 2 diabetes in adulthood. Second, in Chinese traditional society, men had been treated better than women in postnatal life. For example, men had the priority of accessing nutrition food, and women might even be abandoned after birth. Also, women may still be in an undernutritioned status in childhood and had more chances to increase the risk of type 2 diabetes in adulthood than men [14]. Third, the DNA methylation alteration after exposure to prenatal famine was sex-specific [37].

The present study has several advantages. First, the present study examined the relationship between the Chinese famine effects and dysglycemia among participants who were about 50 years older when the 2010–2012 CNNHS was conducted. Generally, the probability of developing type 2 diabetes aged older than 45 was higher than the younger age group, and it was reasonable to detect the relationship among our selected participants [38]. Second, the data of this study were from the large national study, which had the standard criteria of the stratified sampling and covered 30 provinces or autonomous regions. Therefore, the participants of this study were representative of the population who were birth in famine and after the famine in China. Third, compared with other country famine studies, the Chinese famine lasted much longer (three years) and affected more people, which provided us with the opportunity to test the developmental origins hypothesis. Nevertheless, the present study also has some limitations that should be mentioned. First, although we considered smoking, drinking, and diet, etc. as confounding factors, other confounding factors were not considered completely in this study; this may be a limitation. Second, the Chinese famine might leave the stronger and healthier participants due to the excess mortality in early life; This selection bias may decrease the risk of diabetes for the exposed group. Third, the present study did not measure serum insulin, which cannot explain the mediation of insulin secretion on the association of fetal famine with type 2 diabetes. Furthermore, the nutrition intake during infancy and early childhood are also important for diabetes in adulthood. However, this information wasn’t collected because our study is a retrospective study. It is impossible for the participants to recall the information during their infancy and early childhood.

## 5. Conclusions

We found that exposure to famine during fetal life was associated with the risk of diabetes in adulthood. Our results reinforced that very early development is a crucial period for determining the consequence of health in adulthood. The potential mechanisms behind the association need further studies to illuminate. The current study provides an implication for public health. The nutrition status during pregnancy is very important for the health of the whole life. Although famine doesn’t exist in China, in some very poor areas and during some emergency periods, such as earthquakes and floods, pregnant women should be paid more attention to ensure adequate nutrients intake.

## Figures and Tables

**Table 1 ijerph-17-02210-t001:** General characteristics of participants.

Variables	Total	Exposed	Control	*P* Value
Sample size	7830	4081	3749	
Gender				0.409
Male	3323(42.4%)	1750(42.9%)	1573(42.0%)	
Female	4507(57.6%)	2331(57.1%)	2176(58.0%)	
Age	49.7 ± 1.5	50.9 ± 1.0	48.4 ± 0.8	<0.001
Education				<0.001
Primary school and low	2362(30.2%)	1279(31.3%)	1083(28.9%)	
Junior middle school	3047(38.9%)	1415(34.7%)	1632(43.5%)	
High school and above	2421(30.9%)	1387(34.0%)	1034(27.6%)	
Economic				0.037
Low	3711(47.4%)	1961(48.1%)	1750(46.7%)	
Middle	2869(36.6%)	1457(35.7%)	1412(37.7%)	
High	791(10.1%)	439(10.8%)	352(9.4%)	
Smoke				0.117
No	5352(68.4%)	2747(67.3%)	2605(69.5%)	
Yes	2460(31.4%)	1324(32.4%)	1136(30.3%)	
Drinking				0.124
No	4927(62.9%)	2537(62.2%)	2390(63.8%)	
Yes	2886(36.9%)	1532(37.5%)	1354(36.1%)	
Physical exercise				0.154
No	1804(91.0%)	909(91.7%)	895(90.3%)	
Yes	164(8.3%)	74(7.5%)	90(9.1%)	
Sedentary time	2.0(2.0,3.0)	2.0(2.0,3.0)	2.0(2.0,3.0)	0.234
Whole cereal and beans intake levels				0.013
Insufficient	4724(60.3%)	2413(59.1%)	2311(61.6%)	
Sufficient	565(7.2%)	311(7.6%)	254(6.8%)	
Very sufficient	151(1.9%)	94(2.3%)	57(1.5%)	
Livestock and poultry intake levels				0.441
Insufficient	1881(24.0%)	991(24.3%)	890(23.7%)	
Sufficient	1224(15.6%)	643(15.8%)	581(15.5%)	
Excessive	2335(29.8%)	1184(29.0%)	1151(30.7%)	
Body mass index(kg/m^2^)	24.3 ± 3.4	24.4 ± 3.4	24.3 ± 3.4	0.256
Fasting plasma glucose (mmol/L) ^1^	5.4 ± 1.3	5.4 ± 1.3	5.3 ± 1.2	0.111
Type 2 diabetes				0.016
No	7375(94.2%)	3819(93.6%)	3556(94.9%)	
Yes	455(5.8%)	262(6.4%)	193(5.1%)	
Impaired glucose tolerance				0.544
No	6996(94.9%)	3617(94.7%)	3379(95.0%)	
Yes	379(5.1%)	202(5.3%)	177(5.0%)	
Impaired fasting glucose				0.256
No	6881(93.3%)	3551(93.0%)	3330(93.6%)	
Yes	494(6.7%)	268(7.0%)	226(6.4%)	

Data are presented as mean ± SD for continuous variables and N (%) for categorical variables. *P* values in t test for difference in means or χ^2^ test for the difference in proportions between the exposed and unexposed group. Abbreviation: BMI, body mass index; FPG, fasting plasma glucose; IGT, impaired glucose tolerance; IFG, impaired fasting glucose.^1^ Fasting plasma glucose was calculated among participants expect for who had been diagnosed as type 2 diabetes and took antidiabetic medicine regularly.

**Table 2 ijerph-17-02210-t002:** The associations between fetal famine exposure and the risk of dysglycemia in adulthood.

Variables	Unadjusted	Model 1	Model 2	Model 3	Model 4
Fasting plasma glucose					
β	0.04	0.05	0.05	0.05	0.04
*P* value	0.107	0.104	0.079	0.072	0.118
Type 2 diabetes					
ORs (95%CI)	1.26(1.04,1.53)	1.24(1.02,1.50)	1.25(1.03,1.52)	1.25(1.03,1.52)	1.23(1.01,1.50)
*P* value	0.016	0.030	0.023	0.024	0.042
Impaired glucose tolerance					
ORs (95%CI)	1.07(0.87,1.31)	1.06(0.86,1.31)	1.07(0.87,1.32)	1.07(0.87,1.32)	1.06(0.86,1.31)
*P* value	0.545	0.563	0.531	0.541	0.598
Impaired fasting glucose					
ORs (95%CI)	1.11(0.93,1.34)	1.12(0.93,1.35)	1.10(0.91,1.33)	1.11(0.92,1.33)	1.10(0.91,1.33)
*P* value	0.256	0.232	0.308	0.291	0.320

Data were presented as β for the increasing value of the fasting plasma glucose, Ors (95%CI) for the risk odds ratio (95% confidence interval) of type 2 diabetes, IFG, IGT. Model 1 adjusted for gender, economic status, education level. Model 2 adjusted for the variables in model 1 and physical exercise, sedentary time, smoking, drinking. Model 3 adjusted for the variables in model 2 and dietary factors. Model 4 adjusted for the variables in model 3 and BMI.

**Table 3 ijerph-17-02210-t003:** Associations between fetal famine exposure and the risk of dysglycemia in adulthood in different area.

Variables	Unadjusted	Model 1	Model 2	Model 3	Model 4
Moderate famine affected area					
Fasting plasma glucose					
β	0.00	–0.01	–0.01	–0.01	–0.02
*P* value	0.962	0.724	0.753	0.882	0.607
Type 2 diabetes					
ORs (95%CI)	1.13 (0.89,1.44)	1.11 (0.87,1.41)	1.11 (0.87,1.42)	1.11 (0.87,1.42)	1.08 (0.84,1.39)
*P* value	0.304	0.409	0.403	0.402	0.564
Impaired glucose tolerance					
ORs (95%CI)	1.04 (0.80,1.34)	1.04 (0.80,1.34)	1.03 (0.79,1.34)	1.02 (0.79,1.33)	1.01 (0.77,1.31)
*P* value	0.783	0.795	0.826	0.858	0.959
Impaired fasting glucose					
ORs (95%CI)	1.21 (0.97,1.51)	1.20 (0.96,1.50)	1.18 (0.95,1.48)	1.20 (0.96,1.51)	1.18 (0.94,1.49)
*P* value	0.084	0.101	0.142	0.114	0.154
Sever famine affected area					
Fasting plasma glucose					
β	0.09	0.08	0.09	0.09	0.08
*P* value	0.011	0.012	0.007	0.010	0.014
Type 2 diabetes					
ORs (95%CI)	1.41 (1.13,1.76)	1.40 (1.12,1.74)	1.42 (1.13,1.78)	1.42 (1.13,1.78)	1.40 (1.11,1.76)
*P* value	0.002	0.003	0.002	0.002	0.004
Impaired glucose tolerance					
ORs (95%CI)	1.09 (0.85,1.39)	1.09 (0.85,1.39)	1.10 (0.86,1.42)	1.10 (0.85,1.41)	1.09 (0.85,1.40)
*P* value	0.512	0.516	0.458	0.472	0.511
Impaired fasting glucose					
ORs (95%CI)	0.99 (0.78,1.24)	1.00 (0.80,1.26)	1.00 (0.79,1.26)	0.99 (0.78,1.25)	0.98 (0.78,1.24)
*P* value	0.897	0.999	0.984	0.912	0.889

Data were presented as β for the increasing value of the fasting plasma glucose, Ors (95%CI) for the risk odds ratio (95% confidence interval) of type 2 diabetes, IFG, IGT. Model 1 adjusted for gender, economic status, education level. Model 2 adjusted for the variables in model 1 and physical exercise, sedentary time, smoking, drinking. Model 3 adjusted for the variables in model 2 and dietary factors. Model 4 adjusted for the variables in model 3 and BMI.

**Table 4 ijerph-17-02210-t004:** Associations between fetal famine and the risk of dysglycemia in adulthood in different genders.

Variables	Unadjusted	Model 1	Model 2	Model 3	Model 4
Male					
Fasting plasma glucose					
β	−0.01	−0.01	−0.01	−0.01	0.00
*P* value	0.831	0.868	0.896	0.898	0.952
Type 2 diabetes					
ORs (95%CI)	1.22 (0.93,1.61)	1.20 (0.91,1.59)	1.21 (0.91,1.60)	1.20 (0.91,1.59)	1.20 (0.90,1.59)
*P* value	0.147	0.185	0.183	0.191	0.222
Impaired glucose tolerance					
ORs (95%CI)	1.02 (0.74,1.40)	1.02 (0.74,1.40)	1.03 (0.74,1.42)	1.04 (0.75,1.43)	1.02 (0.74,1.42)
*P* value	0.902	0.906	0.865	0.826	0.895
Impaired fasting glucose					
ORs (95%CI)	0.90 (0.69,1.18)	0.91 (0.69,1.19)	0.88 (0.67,1.16)	0.89 (0.67,1.17)	0.89 (0.68,1.18)
*P* value	0.466	0.496	0.375	0.397	0.429
Female					
Fasting plasma glucose					
β	0.08	0.08	0.09	0.09	0.08
*P* value	0.026	0.025	0.020	0.016	0.037
Type 2 diabetes					
ORs (95%CI)	1.30 (0.99,1.70)	1.28 (0.98,1.68)	1.30 (0.99,1.71)	1.30 (0.99,1.71)	1.26 (0.96,1.66)
*P* value	0.058	0.070	0.060	0.060	0.101
Impaired glucose tolerance					
ORs (95%CI)	1.10 (0.84,1.45)	1.09 (0.83,1.43)	1.10 (0.83,1.44)	1.09 (0.83,1.44)	1.08 (0.82,1.43)
*P* value	0.490	0.551	0.516	0.543	0.580
Impaired fasting glucose					
ORs (95%CI)	1.32 (1.03,1.70)	1.33 (1.03,1.71)	1.32 (1.02,1.70)	1.32 (1.02,1.71)	1.31 (1.01,1.70)
*P* value	0.029	0.028	0.034	0.033	0.040

Data were presented as β for the increasing value of the fasting plasma glucose, Ors (95%CI) for the risk odds ratio (95% confidence interval) of type 2 diabetes, IFG, IGT.Model 1 adjusted for gender, economic status, education level. Model 2 adjusted for the variables in model 1 and physical exercise, sedentary time, smoking, drinking. Model 3 adjusted for the variables in model 2 and dietary factors. Model 4 adjusted for the variables in model 3 and BMI.

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
