# Peer review of "Exposure to Chinese Famine in Fetal Life and the Risk of Dysglycemiain Adulthood"

_ijerph, 2020, doi:10.3390/ijerph17072210_

Round 1

Reviewer 1 Report

The authors have developed an interesting article about the relationship between the famine and later risk of dysglycemiain. They have a large sample of subjects. In general is a good study, well described and with interesting results although not revelled new findings.

I would like to suggest to include some data, or if the authors have not the information, include at the end of the article, in the “limitation” section:

- Do you know the percentage of people with moderate or severe famine? This is an important data
- How the author measure the economic status of participants?
 Do you know the economic status and educational status of parents’ parcipants? Because it could be also a bias during the pregnant period

Author Response

Thank you very much for your valuable comments and suggestions on this article. We replied each comment as the follows, and we also revised the manuscript accordingly.

Point 1: Do you know the percentage of people with moderate or severe famine? This is an important data.

Response 1: Thanks for your valuable comment. A sentence "Among the exposed group, 1896(47.7%) participants were from moderate famine area and 2079(52.3%) participants were from severe famine area" was added in line 81-83.

Point 2: How the author measure the economic status of participants?

Response 2: An explanation was added as "The annual family income per capita was used to assess the economic status of participants, and according to the per capita income of urban and rural households in 2011, economic status was divided into three levels: low income (<20000RMB), middle income (20000-40000RMB) and high income (>40000RMB)". (line 103-106).

Point 3: Do you know the economic status and educational status of parents’ parcipants?

Response 3: It is regretful that we did not collect the economic status and educational status of participants’ parents.

Reviewer 2 Report

Authors of the manuscript entitled „Exposure to Chinese Famine in Fetal Life and the Risk of Dysglycemiain Adulthood” present a very interesting study, but major corrections are necessary.

General

The manuscript is prepared shabbily. There are a lot of editorial problems, that Authors should correct.

Introduction

  • Authors should remove “Global report on diabetes”.
  • Line 37 - the first part of the sentence is incomprehensible.

Materials and methods

  • Instead of Methods section, there should be Materials and methods section.
  • Please specify how glucose measurements were carried out for the project participants. Were the measurements taken at the same time for all patients?
  • Authors should complete the information about the methodology, namely food frequency questionnaire (FFQ) and 24-h recalls, because publication no. 20 is in Chinese, which makes it impossible for a number of readers to get familiar with the exact methodology.
  • Please complete the inclusion criteria for patients regarding body weight and health.
  • Authors should complete also the information about the statistical analysis that was applied. How did Authors verify the normality of distribution of data? Which tests did they use?

Results

  • Please complete the description under the Table 4. Each table should have a description.
  • Some results should be presented in details.

Discussion

References should be corrected.

  • Nutrition during childhood is important for diabetes in adulthood, so it is important if it is known what was the nutrition during infancy, early childhood and adulthood in patients (results from FFQ and 24-recalls are highly recommended)? Authors should analyze nutrition in subsequent periods of life. Please complete this information and discuss it in the manuscript. If authors do not have such data, it is a really major limitation for their study.
  • Authors report nutritional behaviors and insulin secretion as risk factors for type 2 diabetes. Did Authors assess insulin resistance and did authors assess nutritional behaviors in patients? Please complete the results.
  • Authors wrote that “animal evidence show that malnutrition in fetal life might affect the appetite, feeding behavior and adiposity in adulthood”. In in the presented study, patients had normal body weight. Did the authors intentionally recruit participants with normal BMI? What were the inclusion criteria?
  • Authors wrote that “The probable reason of attenuated estimated risk could be explained by that obesity was the intermediate factors on fetal famine and type 2 diabetes in adulthood.”

If this sentence relates to own results, it is unclear because patients had normal body weight.

  • Authors should remove the description of the results from the discussion (lines 164-165).
  • The order of publication 27 and 28 in the manuscript is incorrect.
  • Publications are not cited correctly in the text, for example Li et al., should be Li et al. [23] (lines 170, 171, 182).

References

References must be corrected.

  • WHO Global report on diabetes - please complete city and country.
  • Authors must complete the DOI for their references
  • Authors should follow the instructions for authors (the way of referring is incorrect).

Round 2

Reviewer 2 Report

There are still a lot of editorial problems, that Authors did not correct. For example: Baker DJ [8-9]; there is no space in line 48; 2,1 Study Design; International Diabetes Federation (IDF) 1999 on Diabetes 112 Mellitus was used[19-20]; and other.

Please remove from the abstract: "Background:,Methods:" et cetera

Point 9 - If the data from FFQ and 24-recall are not presented, this information should be provided in the text. Please complete this.

Point 10 - Please include this information as a study limitation.

In my opinion, the limitations of presented study are so important that they should be also presented in the Conclusion section.

Author Response

Response to Reviewer 2 Comments

Thank you very much for your valuable comments and suggestions on this article. We replied each comment as the follows, and we also revised the manuscript accordingly.

Point 1: There are still a lot of editorial problems, that Authors did not correct. For example: Baker DJ [8-9]; there is no space in line 48; 2,1 Study Design; International Diabetes Federation (IDF) 1999 on Diabetes 112 Mellitus was used[19-20]; and other.

Response 1: Thank you for reminding. We carefully checked the editorial problems of the full text and made corresponding modifications.

Point 2: Please remove from the abstract: "Background:,Methods:" et cetera

Response 2: We revised the abstract section according to the format of the Journal and removed from the abstract: "Background:,Methods:, Results:, Conclusions:” (line9,11,16,22)

Point 3: If the data from FFQ and 24-recall are not presented, this information should be provided in the text. Please complete this.

Response 3: Previous studies showed clear associations of  whole cereal, beans, livestock  and poultry intakes with the diabetes. Therefore, the present study involved  the intake of whole cereal and beans and the intake of livestock and poultry as confounders when exploring the  association of famine exposure in fetal life with dysglycemia in adulthood, and didn't describe the whole dietary intake patterns. We added these description  in Data collection section as : "Previous studies had indicated associations of whole cereal, beans, livestock  and poultry intakes with the diabetes [18-20]. Therefore, the present study involved  the intake of whole cereal and beans and the intake of livestock and poultry as confounders, and the other information of dietary intake was not presented." (line 120-123)

Point 4: Point 10 - Please include this information as a study limitation.

Response 4: We added this as a limitation: "Third, the present study did not measure serum insulin, which cannot explain the mediation of insulin secretion on the association of fetal famine with the type 2 diabetes" (line 291-293 )